# In Situ Identification of Both IL-4 and IL-10 Cytokine–Receptor Interactions during Tissue Regeneration

**DOI:** 10.3390/cells12111522

**Published:** 2023-05-31

**Authors:** Krisztina Nikovics, Anne-Laure Favier, Mathilde Rocher, Céline Mayinga, Johanna Gomez, Frédérique Dufour-Gaume, Diane Riccobono

**Affiliations:** 1Imagery Unit, Department of Platforms and Technology Research, French Armed Forces Biomedical Research Institute, 91223 Brétigny-sur-Orge, France; anne-laure.favier@intradef.gouv.fr (A.-L.F.); mrocher989@gmail.com (M.R.); celine.mayinga@supbiotech.fr (C.M.); johanna.gomes@supbiotech.fr (J.G.); 2War Traumatology Unit, Department of NRBC Defense, French Armed Forces Biomedical Research Institute, 91223 Brétigny-sur-Orge, France; frederique.dufour-gaume@def.gouv.fr; 3Department of Radiation Bioeffects, French Armed Forces Biomedical Research Institute, 1, Place du Général Valérie André, 91223 Brétigny-sur-Orge, France; diane.riccobono@intradef.gouv.fr

**Keywords:** IL-4, IL-10, receptor, interaction, regeneration, proximity ligation assay

## Abstract

Cytokines secreted by individual immune cells regulate tissue regeneration and allow communication between various cell types. Cytokines bind to cognate receptors and trigger the healing process. Determining the orchestration of cytokine interactions with their receptors on their cellular targets is essential to fully understanding the process of inflammation and tissue regeneration. To this end, we have investigated the interactions of Interleukin-4 cytokine (IL-4)/Interleukin-4 cytokine receptor (IL-4R) and Interleukin-10 cytokine (IL-10)/Interleukin-10 cytokine receptor (IL-10R) using in situ Proximity Ligation Assays in a regenerative model of skin, muscle and lung tissues in the mini-pig. The pattern of protein–protein interactions was distinct for the two cytokines. IL-4 bound predominantly to receptors on macrophages and endothelial cells around the blood vessels while the target cells of IL-10 were mainly receptors on muscle cells. Our results show that in situ studies of cytokine–receptor interactions can unravel the fine details of the mechanism of action of cytokines.

## 1. Introduction

The understanding of the mechanisms of inflammation and tissue regeneration after wounding (irradiation, traumatic injury, and infection) is of great scientific and clinical interest. Immune cells form heterogeneous cell populations that acquire functional specialization in response to changes in the microenvironment [1,2,3]. Immune cells have long been known to induce inflammation and coordinate the efficient repair of damaged tissues [4,5,6]. However, the molecular and cellular mechanisms by which they exert their effects remain to be elucidated. If regeneration is not coordinated, fibrosis can occur [7,8]. Therefore, studying immune cell communication using different biological, molecular, and cellular approaches is important for the development of therapeutic strategies to repair tissues.

Cytokines secreted by immune cells have been shown to enhance tissue regeneration. Cytokines can be pro- or anti-inflammatory cytokines. Pro-inflammatory cytokines (Interleukin (IL)-1β, Tumor Necrosis Factor (TNF), and Interferon-gamma (IFN-γ)) promote the progression of inflammation in the early stages of inflammation. In contrast, anti-inflammatory cytokines (IL-4, IL-10, and IL-13) play a role in inhibiting inflammatory processes and promoting tissue regeneration [9,10]. IL-4, discovered in 1981, is a secreted cytokine [11,12]. CD4 T-cells, basophils, eosinophils, mast cells, NK T- and innate lymphoid cells 2 (ILC2) are the main IL-4 producers (Figure 1A, Table 1) [13,14,15,16,17,18,19].

Cytokine receptor-bearing cells respond to cytokines. The IL-4R is ubiquitously expressed and has been detected at the surface of B cells, T cells, mast cells, eosinophils, basophils, monocytes, macrophages, and non-lymphohematopoietic cells (fibroblasts, endothelial cells, airway epithelial cells, smooth muscle cells, and keratinocytes) (Figure 1, Table 1) [19,20,21,22]. IL-13 also binds to the IL-4R and triggers a different signaling pathway [20].

IL-10 was discovered 30 years ago by Fiorentino and colleagues (1989) [23]. They showed that IL-10 was secreted by T-helper Th2 cell clones and inhibited cytokine production by Th1 cells [23]. IL-10 was also found to be secreted by CD4 and CD8 T cells, B cells, [24] macrophages, monocytes, dendritic cells (DC), neutrophils, mast cells, eosinophils, and natural killer cells (Figure 1, Table 1) [25,26,27,28]. In addition, some non-hematopoietic cells, epithelial cells, and tumor cells could also produce IL-10 [29]. IL-10 is a multifunctional cytokine; its main function is to suppress inflammatory responses. In addition, it regulates the function of monocytes, macrophages, B cells, NK cells, cytotoxic and helper T cells, mast cells, granulocytes, dendritic cells, keratinocytes, and endothelial cells (Figure 1; Table 1) [25,26,27,28,30]. IL-10 is recognized by its cytokine-specific receptor [31].

Studies have also shown that cytokines act in coordination ensuring the effectiveness of tissue regeneration. Although the mechanism of action of cytokines is not fully unraveled, cytokines are known to act by binding to their cognate receptors (Figure 1). The activation of cells is regulated by the interaction of the receptors with cytokines [32]. Therefore, in animals and other organisms, the localization of cytokines in situ is not sufficient to understand the different steps of inflammation and tissue regeneration.

A special type of immune cell is the macrophage and, more particularly, the tissue-resident macrophage, which, in addition to playing a role in homeostasis plays an important role in the regeneration processes by continuously monitoring internal and external signals [33]. The other large group of macrophages that is located in the injured area is monocyte-derived macrophages [34]. Both tissue-resident and monocyte-derived macrophages can evolve into different phenotypes depending on their mode of activation [35]. This polarization is influenced in vitro by several cytokines (Figure 2A). In 1992, Gordon and coworkers demonstrated that IL-4 initiated M2 macrophage polarization of peritoneal macrophages, in contrast to IFN-γ, which polarized them into M1 macrophages [36,37,38]. In 1995, IL-13 was identified as the cytokine that shares some functionality with IL-4 [39]. Over the years, other M2 macrophage activating factors were described (IL-4/IL-13 for M2a, IL-1R agonists/Toll-like receptor for M2b, IL-10 for M2c, and IL-6 for M2d) [9]. However, the mechanism of polarization in vivo is not well understood (Figure 2B) [40]. M1 macrophages and M2 macrophages have distinct roles in tissue regeneration [41]. M1 macrophages produce pro-inflammatory cytokines involved in tissue clearing to remove pathogens through phagocytosis, which would otherwise maintain inflammation [42]. The tissue regeneration phase is characterized by the change in phenotype of M1 macrophages into M2 macrophages that increase in number by producing anti-inflammatory cytokines, such as IL-10 and TGF-β [43]. M2 polarization of macrophages up-regulates the expression of the mannose receptor (CD206) [36,37]. In situ, the differentiation between tissue-resident and M2/M2-like macrophages is quite complicated because both express the CD206 marker [3,44].

Several methods exist already to study protein–protein interactions in vivo with advantages and limitations: (i) Yeast two-hybrid systems are often used to identify protein–protein interactions in vivo [45]. However, the post-transcriptional modifications of some proteins may differ in yeast from those of animal cells, even if it is a eukaryote system. Moreover, it is not adapted to detect receptor-binding proteins. Therefore, protein–protein interactions observed by such methods may differ from the actual in situ interactions. (ii) Forster resonance energy transfer (FRET) is used to study in situ the protein–protein interactions [46]. However, this technique is somewhat cumbersome due to the autofluorescence in the cells. (iii) Bimolecular fluorescence complementation (BiFC) is another very efficient technique, but it also requires chimeric proteins for analysis [47]. (iv) Therefore, we were interested in a new innovative method, the proximity ligation assay (PLA) that allows the in situ detection of specific interaction events within protein complexes. In this technique, a couple of designed antibodies (each antibody is conjugated to oligonucleotides) bind in situ to two target proteins, suspected to interact with each other [48,49,50,51,52]. The oligonucleotides, through a polymerase-mediated rolling circle reaction, can be amplified and labeled with a fluorophore. The resulting fluorescence confirms the interaction between two proteins of interest. The advantage of the PLA method is the higher sensitivity and specificity compared with the FRET technique [32]. Although PLA has many advantages, one of its major disadvantages is its dependence on enzymes, which makes the method expensive and imposes requirements on enzyme storage and stability.

Our work aims to investigate the in situ interaction of two key anti-inflammatory cytokines (IL-4 and IL-10) and their receptors during tissue regeneration using a proximity ligation assay (PLA) (Figure 1B). Currently, little information is available on mammalian M2 macrophages, other than rodents and humans. The pig has proved to be an excellent model for understanding human macrophages, as tissue regeneration in the pig is very close to that in humans [53,54,55]. Therefore, three inflammatory pig models were studied to determine whether the observed interactions were general or wound-specific: skin tissue from an injured ear, an infected lung, and irradiated skeletal muscle tissue, respectively. The challenging detection of the cytokine–receptor interaction in situ was a success that could provide new insights into cellular activation during tissue regeneration.

## 2. Materials and Methods

### 2.1. Tissue Samples

This study was approved by the French Army Animal Ethics Committee (N°2011/22.1). All pigs were treated in compliance with the European legislation (dir 2010/63/EU) implemented into French law (decree 2013-118) regulating animal experimentation. Three kinds of tissues were selected: (i) Skeletal muscle tissue from mini-pigs irradiated locally in the lumbar region with 60 cobalt sources at a dose rate of 0.6 Gy/min up to 50 Gy in the entry area. The irradiated muscles were harvested on day 76 after irradiation [56]. (ii) Lung tissue was collected from an animal with several signs of lower respiratory tract infection. (iii) Two days after the injury, skin tissue was collected from the ear that received a small voluntary cut.

### 2.2. Histological Staining

Histological staining was performed as described in [57]. Hematoxylin, phloxine, and saffron (HPS) staining were performed as follows: sections were embedded in 40 s hemalum (11,487, Merck, Darmstadt, Germany) buffer (0.2 g hemalum, 5 g aluminum potassium sulfate in 100 mL distilled water), 3 min in water, 30 s in phloxine (15,926, Merck, Darmstadt, Germany) buffer (0.5 g phloxine in 100 mL distilled water), 1 min in water, 2 min in 70% ethanol, 30 s in 95% ethanol, 1 min in 100% ethanol and 1 min in 100% ethanol. The samples were put in a saffron buffer for 10 min (6 g saffron (S8381, Sigma, Lezennes, France) in 200 mL absolute ethanol) and rinsed with absolute ethanol. Finally, the nuclei were dyed blue, the cytoplasm pink, and the tissue conjunctive orange.

### 2.3. Immunolabeling

Two types of immunolabeling were performed, one by enzymatic reaction (horseradish peroxidase) and the other by fluorescence detection. For both methods, muscle, skin, and lung tissues were harvested immediately after sacrifice and immersed in a fixative solution ANTIGENFIX (P0014, DiaPath, Martinengo, Italy) for 24 h at 4 °C. After three washes with PBS (phosphate-buffered saline without Ca and Mg, GAUPBS0001, Eurobio, Les Ulis, France), samples were embedded in paraffin (39602004, Paraplast plus, Leica, Stuttgart, Germany). Sections of 5 μm were cut from the samples using a microtome (HistoCore, MulticutR, Leica, Stuttgart, Germany). The deparaffinization was carried out by successive 5 min washes (twice in 100% xylene, two times in 100% ethanol, two times in 95% ethanol, then in water). To restore the antigenicity of the proteins, sections were incubated with 10 mM citrate (pH:6.0) (AP0533-500, L-Recovery, Aptum, Boulogne-Billancourt, France) in a steam oven for 15 min. After three washes in PBS, the sections were permeabilized for 15 min with 0.5% (*v*/*v*) Triton X100 (112298, Merck, Darmstadt, Germany) buffered with PBS. After three washes with PBS, non-specific binding was blocked with Emerald Antibody Diluent (936B-08, Sigma, Lezennes, France) for 1 h. Then, sections were incubated overnight at 4 °C with the primary antibody as follows: the primary goat anti-CD206 (C20) (sc-34577, Santa Cruz Bio., Heidelberg, Germany) at 1:500 dilution; primary rabbit anti-IL-4R (ab203398, Abcam, Amsterdam, The Netherlands) at 1:200 dilution; primary mouse anti-IL-4 (ab239508, Abcam, Amsterdam, The Netherlands) at 1:500 dilution; primary rabbit anti-IL-10R (ab225530, Abcam, Amsterdam, The Netherlands) at 1:500 dilution, and primary mouse anti-IL-10 (ab25075, Abcam, Amsterdam, The Netherlands) at 1:200 dilution. The sections were washed three times in PBS for 10 min.

In one part, sections for enzymatic immunolabeling were treated with a 10-fold dilution of H_2_O_2_ (H1009, Sigma-Aldrich, Lezennes, France) for 20 min to inhibit endogenous hydrogen peroxidase. After three PBS washes, the slides were incubated with ready-to-use anti-goat IgG horseradish peroxidase reagent (ImmPRESS MP-7405, Eurobio, Les Ulis, France) and counterstained with 3,3′-diaminobenzidine (DAB) (SK-4100, Eurobio, Les Ulis, France) and hematoxylin. Horseradish peroxidase oxidizes DAB, turning it brown and detectable, thus allowing the detection of macrophages. Labeling was detected with a DM6000 epifluorescence microscope (Leica, Stuttgart, Germany) equipped with a monochrome and color digital camera.

In a second part, sections for fluorescent immunolabeling were incubated with a 1:1000 dilution of Alexa Fluor 488 (A-21206, Thermo Scientific, Villebon sur Yvette, France) anti-rabbit secondary antibody and a 1:1000 dilution of Alexa Fluor 568 (ab175704, Abcam, Amsterdam, The Netherlands) anti-goat secondary antibody for 2 h at room temperature. Finally, tissue sections were washed three times in PBS for 10 min and subsequently mounted with Fluoroshield mounting medium supplemented with DAPI (ab104139, Abcam, Amsterdam, The Netherlands). Fluorescence was detected with a confocal microscope (LSM700, Zeiss, Dresden, Germany).

### 2.4. Proximity Ligation Assay (PLA)

The sections were fixed, antigen retrieved and permeabilized as described in the immunolabeling section. To detect protein–protein interactions, the Duolink^®^ in situ red starter kit Mouse/Rabbit (DUO92101-1KT, Sigma-Aldrich, Lezennes, France) was used according to the company’s recommendations. Briefly, the samples were blocked with Duolink^®^ blocking solution for 60 min at 37 °C. Then, sections were incubated overnight at 4 °C with the primary antibody as follows: primary rabbit anti-IL-4R (ab203398, Abcam, Amsterdam, The Netherlands) at 1:200 dilution; primary mouse anti-IL-4 (ab239508, Abcam, Amsterdam, The Netherlands) at 1:500 dilution; primary rabbit anti-IL-10R (ab225530, Abcam, Amsterdam, The Netherlands) at 1:500 dilution, and primary mouse anti-IL-10 (ab25075, Abcam, Amsterdam, The Netherlands) at 1:200 dilution. Primary antibodies were diluted with Duolink^®^ antibody diluent. The sections were washed three times in wash buffer A for 10 min. The secondary antibodies used are bound to DNA oligonucleotides. These antibodies are called PLA probes. One of the two probes is PLUS (+) and the other is MINUS (−). The sections were then incubated with Duolink^®^ plus and minus PLA probes for 60 min at 37 °C. DNA oligonucleotides linked to the secondary antibody were ligated for 30 min at 37 °C (Duolink^®^ ligase) into circular DNA using hybridizing connector oligonucleotides. The sections were washed three times for 5 min in wash buffer A at room temperature. The circular DNA was amplified by Duolink^®^ polymerase at 37 °C for 100 min. Finally, tissue sections were washed two times in wash buffer B for 10 min and subsequently mounted with Duolink^®^ mounting medium supplemented with DAPI. Fluorescence was detected with a confocal microscope (LSM700, Zeiss, Dresden, Germany).

### 2.5. Quantification and Statistical Analysis

Quantitative studies were based on random immunofluorescence assays, analyzing 3 times 1000 cells in different tissues.

All statistical calculations were performed with a one-way analysis of variances (ANOVA) using GraphPad PrismVersion 4.0 [58]. Post-test comparisons, performed only if *p* < 0.05, were made using Bonferroni’s Multiple Comparison Test. Chi 2 (Microsoft Excel).

## 3. Results

### 3.1. Tissue Wounds Induce Endothelial Cell Activation and Immune Cell Infiltration

Rodents are commonly used as animal models to study tissue regeneration, but little information is available for other animal models. Interestingly, tissue regeneration in humans is more similar to that in pigs than it is to that in rodents [53,54,55,59]. Therefore, three tissues from a pig animal model were selected to perform this study (skin, lung, and muscle) and histological staining was used to characterize the morphology (Figure 3).

Strong endothelial cell activation and intense immune cell infiltration were observed in the wounded tissues (Figure 3B,D,F) compared to healthy tissues (Figure 3A,C,E). The infiltration was mainly localized around blood vessels (Figure 3B,D,F). In the skin and lungs, many granulocytes were observed in the lumen of the vessels (Figure 3B,D). In contrast, no granulocytes were observed in the lumen of the vessels or other parts of the wounded-muscle tissue (Figure 3).

### 3.2. Tissue Damage Results in the Accumulation of M2 Macrophages around the Blood Vessel

As histological staining only assumes the presence of macrophages based on cell morphology, immunolabeling with macrophage-specific antibodies is required for their identification.

First, the CD206 marker, highly expressed by the M2 phenotype of macrophages in injured tissues, was used to check the activation of M2 macrophages (Figure 4) [36,37]. The specificity of the anti-CD206 antibody was validated by the lack of signal obtained in the negative control (Figure 4A,C,E,G,I,K).

Since immunolabeling does not allow distinction of macrophage types, cells expressing the CD206 marker in healthy tissues were assumed to be tissue-specific or tissue-resident macrophages, which are always present but not activated (Figure 4B,F,J) [33,34,60]. Indeed, in healthy tissue, macrophage polarization does not occur (Figure 4B,F,J). However, the expression of CD206 in treated tissue may label M2-like macrophages which are highly activated during regeneration (Figure 4D,H,L).

Overall, damaged tissue generates a higher activation of macrophages compared to healthy tissue, as there are more CD206-positive (CD206+) cells in damaged tissue than in healthy tissue. This is particularly evident around blood vessels. In addition, the sphericity of the endothelial cell nuclei showed that the tissues have been exposed to inflammation and the endothelial cells have been activated (Figure 4D,H,L).

### 3.3. Tissue Injury Activates IL-4R Expression on the Surface of Macrophages

Recently, the IL-4 cytokine was the focus of attention because of its key role in tissue regeneration [61]. Under in vitro conditions, IL-4 cytokine can be used to polarize M2a macrophages from monocytes [62]. In a second step, we examined whether tissue injury activates IL-4R expression on the surface of macrophages under in vivo conditions. For this experiment, the CD206/IL-4R co-localization was investigated in the wounded skin, muscle, and lung tissues, with the CD206 marker labeled in red, the IL-4R labeled in green, and the nuclei labeled in blue. The co-localization of both IL-4R and CD206 proteins was studied by immunofluorescence to obtain double labeling on the same tissue sample. In the absence of a primary antibody (negative control), no expression of these markers could be observed (Appendix A).

In the skin sample, (Figure 5A,B), mainly two types of cells were observed, macrophages (CD206+/IL-4R+ cells) (thin arrow) and endothelial cells (CD206-/IL-4R+ cells) (arrowhead) expressing the IL-4R in the internal surface of the blood vessel.

For the muscle sections, similar observations could be made (Figure 5E,F), with slightly higher levels of M2-like macrophages than in the skin sample.

Similarly, a significant number of macrophages expressing IL-4R protein (Figure 5I,J) were identified in the wounded lung. In the lumen of the blood vessels, immune cells showed strong IL-4R expression but were CD206 negative (Figure 5K). No granulocytes were observed since the latter does not express IL-4R.

In summary, the three different injured tissues showed a very similar pattern. Indeed, there were significantly more M2-like macrophages expressing the IL-4R (CD206+/IL-4R+) in wounded tissue (6.3% (skin); 11.4% (muscle); 9.1% (lung)) than in healthy tissues (1.6% (skin); 0.5% (muscle); 1.8% (lung) (Figure 5C,D,G,H,L,M)). Simultaneously, the number of M2-like macrophages without IL-4R (CD206+/IL-4R-) was significantly reduced (2.6% (skin); 7% (muscle); 2.5% (lung)) in wounded tissues compared to healthy tissues (8.5% (skin); 8.5% (muscle); 6% (lung)) (Figure 5C,D,G,H,L,M). Since M2-like macrophages express the IL-4R upon injury, it can be assumed that they are the main targets of the IL-4 cytokine. Moreover, other target cells were also identified during regeneration since an increase in CD206-/IL-4R + cells was detected in wounded tissues (10.5% (skin); 5% (muscle); 5.8% (lung)) compared to the healthy tissue (7.7% (skin); 1.4% (muscle); 3.4% (lung) (Figure 5A,E,I).

### 3.4. Injury in Muscle Tissue Displayed High IL-10R Expression

IL-10 has been shown to play a key role in tissue regeneration by activating M2 macrophages and downregulating the production of pro-inflammatory cytokines. In the next experiment, we investigated the IL-10R expression on the surface of macrophages in vivo. The co-localization of IL-10R and CD206 proteins was studied by immunofluorescence (Figure 6), CD206/IL-10R co-localization, labeling the CD206 marker in red, the IL-10R in green and the nuclei in blue. In the absence of a primary antibody (negative control), no expression of these markers was observed (Appendix A). The number of IL-10R-expressing M2-like macrophages (CD206+/IL-10R+) was increased in wounded tissues (1.4% (skin); 11.3% (muscle)) compared to healthy tissues (0.2% (skin); 0.2% (muscle)) (Figure 6 C,D,G,H,M,N).

The number of M2-like macrophages without IL-10R (CD206+/IL-4R-) was reduced (7.7% (skin); 6.1% (muscle); 4.8% (lung)) in wounded tissues compared to healthy tissues (8.1% (skin); 7% (muscle); 6% (lung) (Figure 6C,D,G,H,M,N)). Moreover, other target cells were also present during regeneration as an increase in the number of CD206-/IL-10R+ cells was observed in damaged tissue (4.1% (skin); 11.5% (muscle); 1.9% (lung)) compared to healthy tissues, respectively (2.1% (skin); 0.1% (muscle); 0.1% (lung) (Figure 6C,D,G,H,M,N)). In addition, regenerating muscle cells showed intense IL-10R expression (Figure 6K,L).

### 3.5. Granulocytes Show High IL-4 Expression

The co-localization between IL-4 and IL-4R was studied in skin, lung, and muscle tissues (Figure 7 and Appendix A). IL-4 is labeled in red, its receptor in green, and the nuclei are labeled in blue (Figure 7 and Appendix A). In the absence of a primary antibody (negative control), no expression of these markers was observed (Appendix A).

For the skin tissue, (Figure 7A,B), two types of IL-4+ cells were identified (IL-4+ cells): (i) granulocytes (thin arrow), expressing a high level of IL-4 in the cytoplasm; (ii) cells detected around the vessel, expressing both IL-4 and IL-4R. Because cytokines can simultaneously be secreted and activate the same cell, it was more difficult to recognize IL-4-secreting cells than IL-4-targeted cells (Figure 7A,B).

Regarding the lung (Figure 7C,D), granulocytes and IL-4+/IL-4R+ cells were detected but in lower levels compared to the skin. All around the lumen of the blood vessels (labeled with a star) were cells expressing IL-4. For the muscle section (Figure 7E,F), the co-localization of IL-4 and IL-4R was very intense only around the vessel.

### 3.6. Expression of IL-10 Cytokine in Different Tissues

The co-localization between the IL-10 cytokine and the IL-10R was confirmed (Figure 8 and Appendix A). In the absence of a primary antibody (negative control), no expression of these markers was observed (Appendix A). In skin and lung tissues (Figure 8A–D), two types of cells were observed: IL-10R+ cells (thin arrow) and some IL-10+ cells (thick arrow). Nevertheless, no co-localization of the two labels was observed. Granulocytes (arrowhead) in the lumen of the vessel did not express either IL-10 cytokine or IL-10R (Figure 8C).

In the muscle sample, several cells around the regenerating muscle were found to be IL-10R+ and IL-10+ positive (Figure 8E,F). However, since cytokines are secreted it is difficult to determine which cells are sources and which are targets of cytokines. Only protein–protein (cytokine–cytokine receptor) interactions can answer this question.

### 3.7. IL-4/IL-4R and IL-10/IL-10R Interaction In Situ

Immunostaining is useful to specifically detect proteins while reliable indications regarding protein–protein interactions could be effectively studied by proximity ligation assays in order to identify, via the identified interaction, the target cells involved.

Interactions between the IL-4 cytokine and its respective receptor IL-4R (Figure 9 and Appendix A) and similar IL-10/IL-10R interactions (Figure 10 and Appendix A) were studied in the different sections. In the absence of a primary antibody (negative control), no protein–protein interaction was observed (Appendix A). In samples from healthy animals, cytokine–receptor interactions were detected in a few cells (Appendix A). In contrast, in wounded tissues, cytokine–receptor interactions were enhanced with very distinct patterns for the two cytokines (Figure 9 and Figure 10). The interaction of IL-4/IL-4R was mainly activated around the vasculature (Figure 9) whereas the activation of IL-10/IL-10R was more localized to muscle cells (Figure 10E,F). Few granulocytes were observed in the vessel lumen without signs of interactions between the two proteins (Figure 9D). Some other immune cells showing a positive IL-4/IL-4R interaction were also observed in the lumen (Figure 9D). 

In summary, the three different damaged tissues showed very similar patterns. Specifically, for IL-4, there were significantly more cytokine receptor interactions in the injured tissues (9% (skin); 17.9% (muscle); 11.1% (lung)) than in the healthy tissues (4.5% (skin); 2% (muscle); 5% (lung)) (Figure 11A). Figure 11B shows that fewer cells in both injured and healthy tissues showed IL-10/IL-10R interactions. However, the tendency was similar to that observed for IL-4, where there were also significantly more cytokine receptor interactions in injured tissues (2% (skin); 15% (muscle); 2% (lung)) than in healthy tissues (1.5% (skin); 0.1% (muscle); 0.1% (lung)) (Figure 11B).

## 4. Discussion

Macrophages play an important role in tissue development, homeostasis, and wound healing. Several phenotypes of macrophages were identified and each of them plays a distinct role during the tissue regeneration process through the expression of a panel of cytokines [44,59,63,64], which have diverse roles in each process [43,59,62]. However, the mechanisms of communication between the immune effectors and the macrophages during tissue regeneration are still poorly understood in vivo. Indeed, in vitro, observations could not be readily transposed to in vivo conditions because in vivo mechanisms are more complex [42].

In vitro, the addition of IL-10 to muscle cell cultures can enhance the percentage of myogenin-expressing cells and improve cell fusion [65]. Injection of both anti-inflammatory cytokines (IL-4 and IL-10) into injured tissue can significantly enhance wound healing in different tissues [26,66,67,68]. In addition, M2c macrophages express a high level of both IL-4 and IL-10, suggesting they play a significant role in promoting muscle regeneration [8,37]. In vivo, intramuscular injection of IL-10 DNA is effective in suppressing inflammation by inhibiting the production of proinflammatory cytokines [69,70,71]. In addition, IL-10 can promote myocardial cell regeneration [72] and skeletal muscle cell healing [73]. Myocytes are capable of co-expressing IL-10 and IL-10R, but this interaction has not yet been experimentally demonstrated under in situ conditions [74]. However, it should also be mentioned that these two cytokines can have a negative effect on tissue regeneration, depending on where, when, and in which animal model they were injected, making their use in therapeutic treatment difficult [21,75]. To make progress in this area, a deeper understanding of the IL-4 and IL-10 cytokine–receptor interaction is needed. Cytokine therapy is an intensively developing field and exciting new advances in understanding the mechanism of action of these cytokines are expected in the future [76].

In our work, we aimed to address this question. Granulocytes are known to be a source of IL-4 [77]. We show that granulocytes in the skin and lungs are the primary IL-4 producers. Rare granulocytes were identified in muscle tissue, as this tissue was collected 76 days after irradiation. In these tissues, IL-4 production was mainly localized around the vasculature. For IL-10, many IL-10+ cells were identified in skin, lung, and muscle sections. No IL-10 expression was detected in granulocytes. Identifying cells expressing both IL-4 and IL-10 will be a task in the future.

The expression of cytokine receptors was also analyzed in healthy tissues (skin, lung, and muscle); IL-4R and IL-10R were weakly but ubiquitously present in different cell types, similar to the findings reported in the literature [78,79]. However, in the different regenerating tissues (regardless of the source of injury), the expression of both receptors was intensely increased at the surface of macrophages. These results suggest that similar to the in vitro results, the activity of these two cytokines is essential for the polarization of macrophages in vivo (Figure 2). In the future, to verify whether the cytokine–receptor interactions also occur in macrophage cells, combined protein–protein interaction with the CD206 immunolabeling of macrophages could be performed. It would also be interesting to detect which subtypes of macrophages are most involved in polarization. As classical immunolabeling is not able to distinguish M2 macrophage subtypes in situ, future work will focus on the identification of macrophage subtypes by in situ hybridization [42].

In addition to macrophages, the induction of receptor expression was also observed in other cells and the two receptors showed different patterns. The IL-4R was most intense in endothelial cells in addition to macrophages, while the IL-10R was most intense in regenerating muscle. The immune system is regulated by cells that produce IL-4 and IL-10 and by cells that respond to IL-4 or IL-10. We are only at the beginning of learning how cytokine/cytokine receptor relationships translate into tissue regeneration. Therefore, it is very important to analyze the interaction of cytokines/cytokine receptors using protein–protein interactions.

The interaction between IL-4 and its receptor occurs in situ, in the specific tissue or cellular environment. If IL-4 is present in the tissue or cellular microenvironment, it can bind to the IL-4Rα subunit on the cell surface, leading to the recruitment of the γc subunit. This receptor–ligand binding results in a series of intracellular signaling events, collectively referred to as the IL-4 signaling pathway. This interaction plays a vital role in the immune response, regulating inflammation and promoting the differentiation of certain immune systems [77,80]. Cytokine/receptor PLA results demonstrated that IL-4 in particular showed cytokine/receptor interaction around blood vessels and in the epithelial cells. Once the inflammation has subsided, new tissue formation must begin, for which fibroblasts and endothelial cells are essential, along with macrophages [81,82]. Fibroblasts are involved in tissue remodeling and endothelial cells are responsible for the formation of new vascular networks [81,82]. During the wound-healing process, there is continuous communication between macrophages, fibroblasts, and endothelial cells [83]. If there is any failure in the communication needed for tissue regeneration, regeneration fails. This study suggests that endothelial cells are activated simultaneously with macrophages, as simultaneous activation of the IL-4R on the surface of both cells was observed. Furthermore, endothelial cells showed intense IL-4 cytokine/receptor interactions during skin, lung, and muscle regeneration (Figure 12, Table 2). Two types of molecules can bind to IL-4R: IL-4 itself and IL-13. IL-4 and IL-13 are closely related cytokines and share a significant degree of structural similarity. Both cytokines initiate similar signaling pathways. IL-4 binds with high affinity to IL-4Rα, whereas IL-13 binds with lower affinity. It is important to note that although IL-4 and IL-13 can bind to IL-4Rα in parallel, the binding may induce functional differences. IL-4 is a potent inducer of the Th2 immune response and is involved in allergy and asthma. However, in addition to these functions, IL-13 may also be associated with tissue remodeling and fibrosis [20,21,39,80]. The interaction between IL-13 and IL-4R will be analyzed in the future [20,39]. Different cytokines can induce different pathways and the analysis of both cytokine/receptor interactions could be very important for a better understanding of tissue regeneration.

IL-10 acts by binding to the IL-10R-specific receptor complex. The IL-10/IL-10R interaction is particularly important in modulating immune responses and limiting excessive inflammation. IL-10 has potent anti-inflammatory properties and can inhibit the production of pro-inflammatory cytokines and chemokines. In addition, it plays a role in regulating the activation and function of various immune cells such as macrophages, dendritic cells, and T cells, promoting an anti-inflammatory environment [25,27,69,70]. When IL-10 is present in the tissue or cellular microenvironment, it can bind to the IL-10Rα subunit on the cell surface, leading to the recruitment and association of the IL-10Rβ subunit. This receptor–ligand binding triggers intracellular signaling events that are responsible for IL-10-mediated effects [31]. Using PLA, we have shown that the primary targets of IL-10 are regenerating muscle cells, contrary to the activation observed for IL-4. In the skin and lungs, IL-10 activation was very low whereas, a very high level of activation was observed in regenerating muscle cells.

Both cytokine–receptor interactions were more intense in damaged tissue than in healthy tissue. The percentage of IL-4/IL-4R interactions detected in lung and muscle was approximately equal to the percentage of cells expressing the receptor in both healthy and injured tissues. In skin, however, the percentage of these cytokine–receptor interactions was approximately half the percentage of cells expressing the receptor. Further research is needed to understand this difference. The analysis of the IL10 cytokine is even more complicated. In healthy tissues, the level of cytokine–receptor interaction was approximately equal to the number of cells expressing the receptor, but in damaged tissues, with the exception of lung tissue, fewer IL-10/IL-10R interactions were detected than the number of cells expressing the receptor. In addition, only a few cytokine–receptor interactions per cell were observed for both cytokines in PLA assays. This phenomenon can be explained by the fact that the duration of the interaction between different cytokines and its receptor can vary depending on several factors, including the concentration of the cytokine, the availability of the receptor at the cell surface, and the downstream signaling events triggered by the interaction. In general, the cytokine–receptor interaction is a transient event and typically lasts for a short period to ensure dynamic and accurate signal transduction. The actual binding time between cytokine and receptor at the molecular level is typically in the range of microseconds to milliseconds [84,85]. However, it is important to note that the total duration of cytokine–receptor interaction and subsequent signal transduction may be longer due to downstream signaling events and internalization of the receptor–ligand complex. Internalization may lead to the cessation of signal transduction and possible degradation or recycling of the receptor–ligand complex [84]. In addition, the low number of cytokine–receptor interactions for IL-4 may be explained by the fact that it may bind to the receptor with another cytokine [21], unlike IL-10, which is the only ligand for the IL-10 receptor [31]. It is important to emphasize that the exact duration of cytokine–receptor interaction and signal transduction is context-dependent and may vary in different cellular and physiological contexts, and further research is needed to fully understand the exact temporal aspects of the interaction.

## 5. Conclusions

In summary, we can conclude that:Using immunolabeling, M2-like macrophages were identified in regenerating tissues.Cut-infection-irradiation injuries induced the expression of IL-4 and IL-10 receptors in M2-like macrophages in all three tissues.Cytokine receptor induction was not limited to macrophages, as other cells also showed receptor induction.The two different cytokine receptors showed different localization. IL-4R expression was mainly expressed in endothelial cells and localized around the vasculature. IL-10R expression was particularly intense in the regenerating irradiated muscle cells.The PLA method provided valuable information for a better understanding of tissue regeneration. Using this technique, we have shown that IL-4 can interact with its receptor around the vasculature and IL-10 interacts with its receptor in regenerating muscle.Granulocytes do not have an IL-4 cytokine/receptor interaction, suggesting that these cells are not targeted but are a source of cytokine.

These results validate that the strategy implemented to detect those interactions in situ is functional.

The relationship between IL-4, IL-10, and tissue regeneration is an active area of research [4,6,7,19,61]. However, the precise role of cytokine–receptor interactions and signal transduction in tissue regeneration is not yet fully understood. These interactions in different tissues are highly context-dependent, influenced by specific individual host factors. The timing and magnitude of IL-4 and IL-10 production, as well as the balance between anti-inflammatory and immunosuppressive effects, are crucial in determining the outcome of tissue healing [22,26,66]. However, the specific mechanisms and precise role of IL-4 and IL-10 in tissue regeneration require further investigation.

## Figures and Tables

**Figure 1 cells-12-01522-f001:**
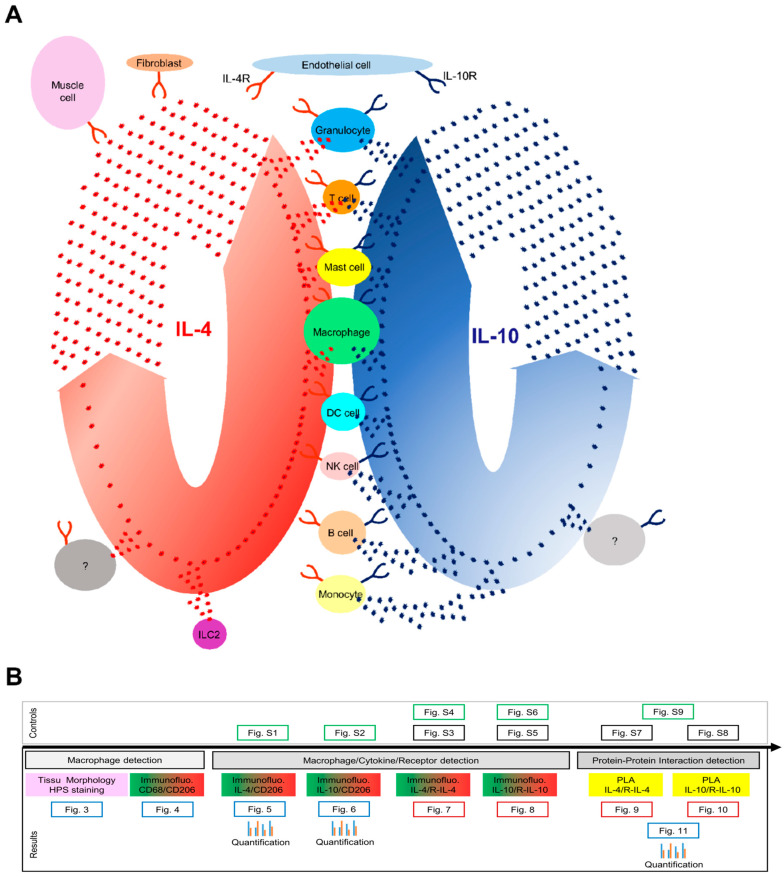
(**A**) Interleukin-4 (IL-4) and interleukin-10 (IL-10) are cytokines secreted by different cells that play an important role in the regulation of immune responses. They have specific receptors through which they exert their effects. Below, we briefly review the cells expressing IL-4 (red asterisks) and IL-10 (blue asterisks) and their receptors IL-4R (red receptor) and IL-10R (blue receptor). There are four types of cells (granulocytes, T cells, mast cells, and macrophages) that produce both cytokines and their receptors. Four cell types (DC cells, NK cells, B cells, and monocytes) secrete only IL-10, but both receptors can be detected on the surface. ILC2 cells express IL-4. The IL-4 receptor is located on the surface of fibroblasts and muscle cells, while both receptors are present on the surface of endothelial cells. In addition, there are cells (marked with a question mark) that produce these cytokines and their receptors, but their identity has not yet been determined. (**B**) Strategy to develop and validate a protein–protein interaction using the PLA method. First, macrophage detection was performed on the tissue of interest; second, actors were identified by immunofluorescence before performing the PLA method to detect protein–protein interaction. Blue boxes represent figures that contain both wounded and healthy tissue. The brown boxes correspond to wounded tissue and the black boxes to healthy tissue. Green boxes indicate negative controls.

**Figure 2 cells-12-01522-f002:**
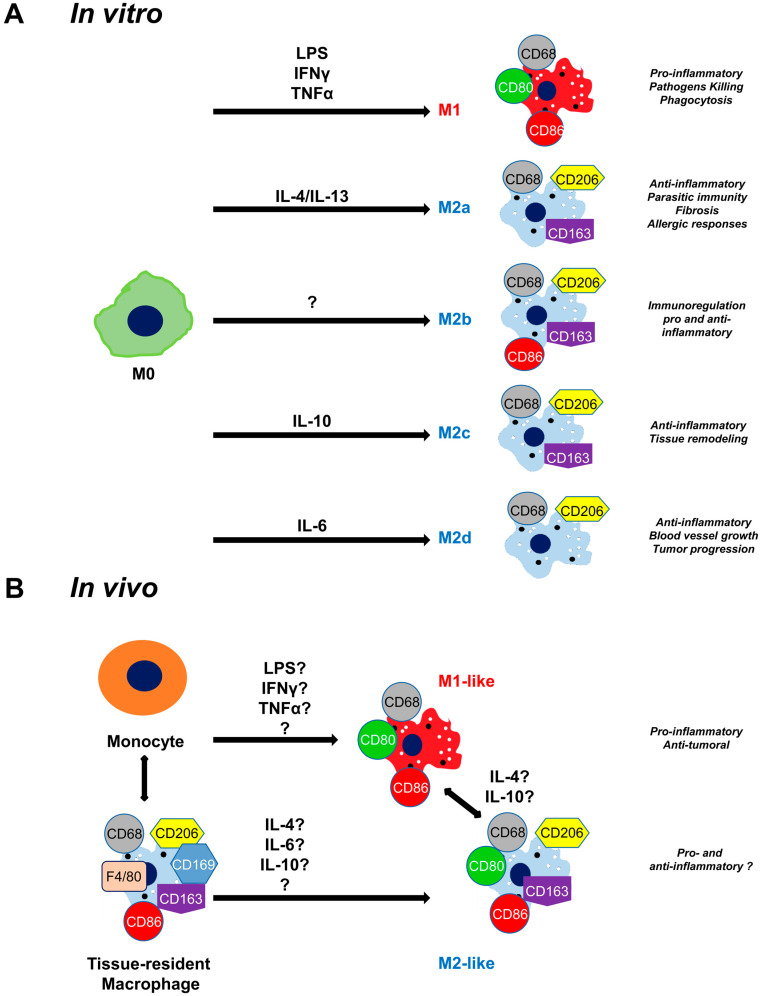
Activation and classification of macrophages: in vitro and in vivo macrophage polarization depending on the activation pathway. (**A**) In vitro, M0 macrophages can be transformed into different phenotypes depending on the activation pathway. This polarization is influenced by different molecules (including several cytokines). IL-4/IL-13 induces polarization of M2a, IL-1R agonists/Toll-like receptor induces polarization of M2b, IL-10 induces polarization of M2c, IL-6 induces polarization of M2d macrophages, in contrast to IFN-γ, TNF, and Lipopolysaccharide (LPS), which induce polarization of M1 macrophages. Understanding the details of polarization is essential as different subtypes have different functions. (**B**) In contrast, the polarization of macrophages in vivo is relatively poorly understood. Further studies are needed to determine which molecules are important in the polarization of the macrophage types involved. M0: non-activated macrophage.

**Figure 3 cells-12-01522-f003:**
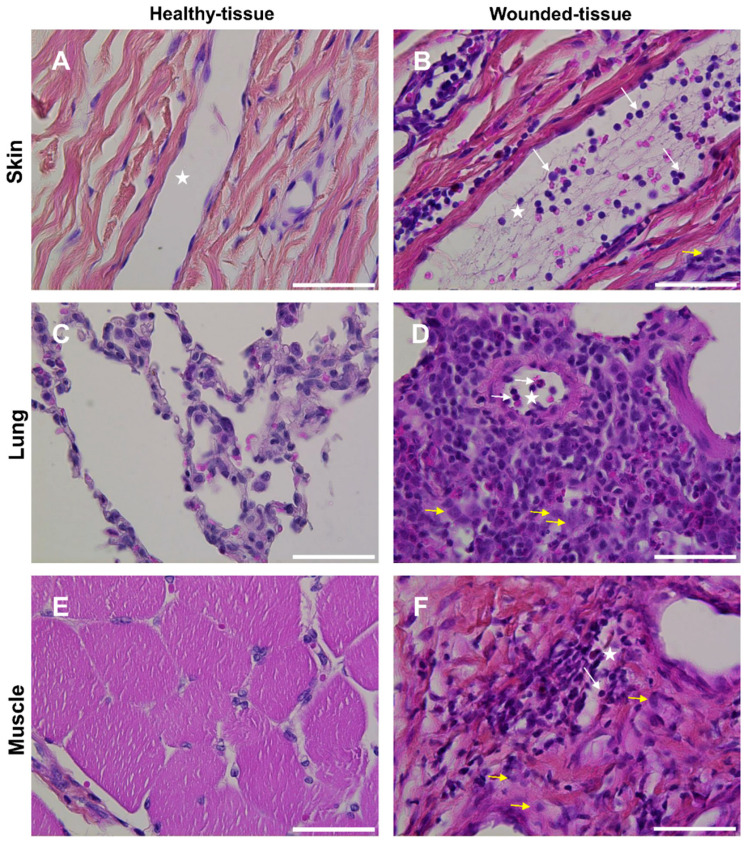
Macroscopic morphology of tissues. Histologic HPS staining was performed on (**A**,**B**) skin (**C**,**D**) lung, and (**E**,**F**) muscle sections in (**A**,**C**,**E**) healthy tissue and (**B**,**D**,**F**) wounded tissue. Scale bar = 50 µm. White arrow: granulocyte; yellow arrow: macrophage; star: lumen of the blood vessel.

**Figure 4 cells-12-01522-f004:**
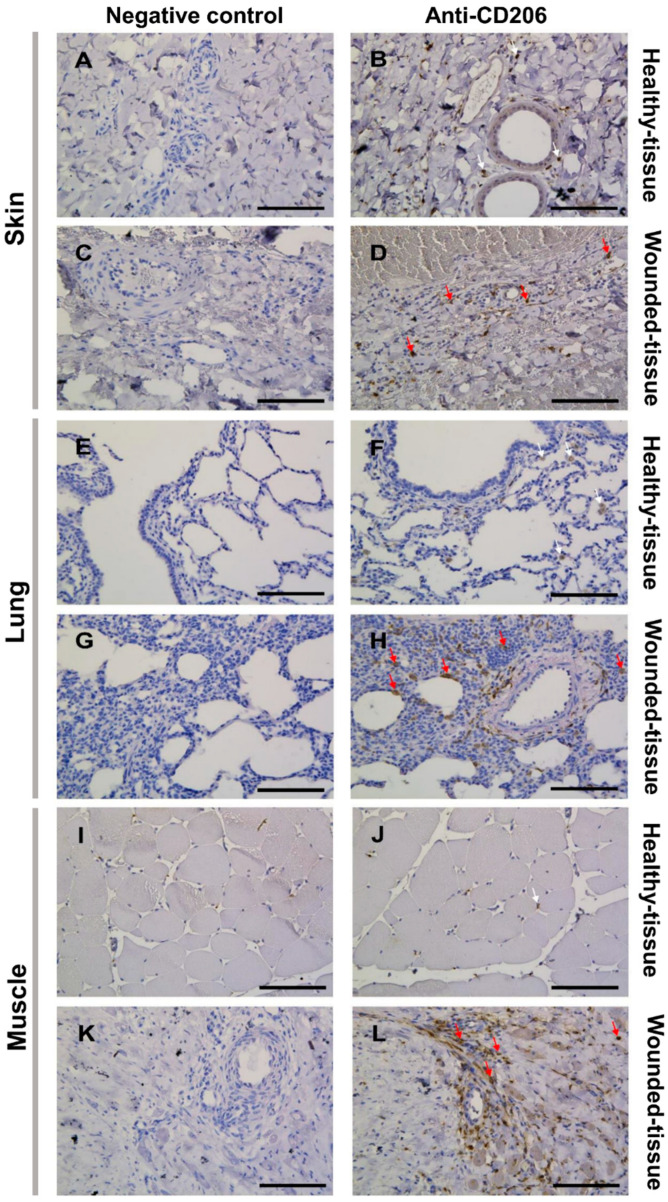
Identification of tissue-resident and M2-like macrophages in situ. (**B**,**D**,**F**,**H**,**J**,**L**) Immunolabeling with CD206 of the (**A**–**D**) skin of the (**E**–**H**) lung and the (**I**–**L**) muscle sections. (**A**,**B**,**E**,**F**,**I**,**J**) Healthy tissue. (**C**,**D**,**G**,**H**,**K**,**L**) Wounded tissue. (**A,C,E,G,I,K**) Negative controls of the immunolabeling. (**B**,**D**,**F**,**H**,**J**,**L**) Anti-CD206 (brown color) labeling the tissue-resident and M2-like macrophages. Nuclear staining with hematoxylin (blue color). Scale bar = 100 µm. White arrow: Tissue-resident macrophage; red arrow: M2-like macrophage.

**Figure 5 cells-12-01522-f005:**
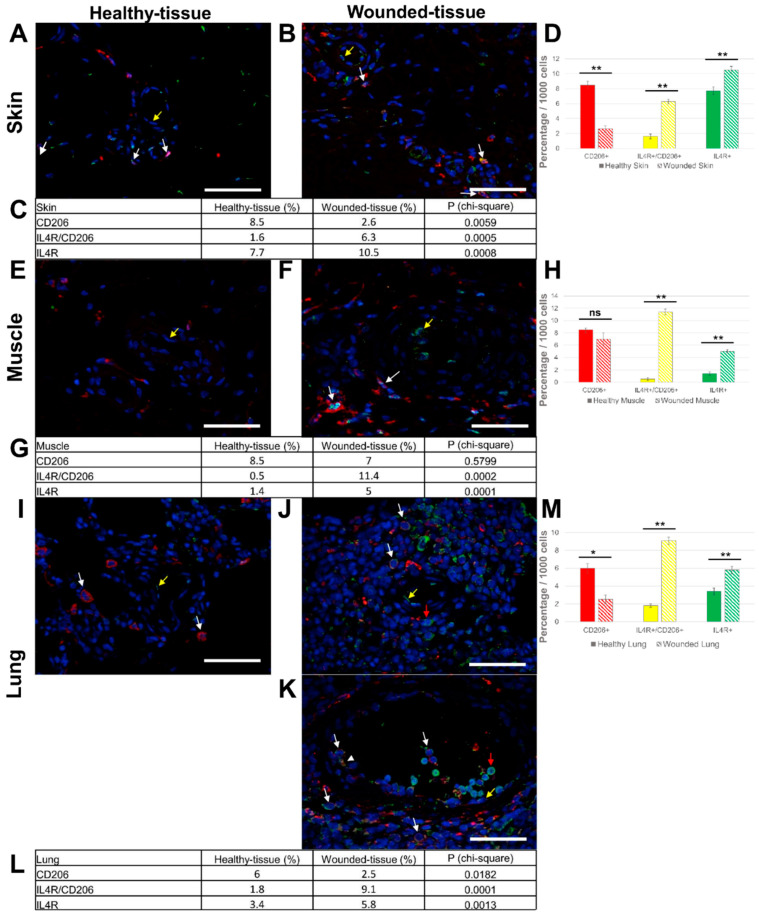
Identification of IL-4R+ and CD206+ cells in situ. (**A**,**B**) Immunolabeling of both IL-4R and CD206 in the skin, (**E**,**F**) the muscle, and (**I**–**K**) the lung sections. (**A**,**E**,**I**) Healthy tissue. (**B**,**F**,**J**,**K**) Wounded tissue. (**C**,**D**) Quantitative analysis of both IL-4R+ and CD206+ cells in the skin; (**G**,**H**) the muscle and (**L**,**M**) the lung. Quantitative analysis based on random examination of 3 sets of 1000 cells per condition. Anti-IL-4R (Alexa488, green fluorescence) and Anti-CD206 (Alexa568, red fluorescence) label the tissue-resident and M2-like macrophages. Nuclear staining with DAPI (blue fluorescence). Scale bar = 50 µm. Yellow arrow: endothelial cell; white arrow: macrophage; and red arrow: IL-4R + cell. ** *p* < 0.01, * *p* < 0.5, ns = not significant, compared to healthy tissue.

**Figure 6 cells-12-01522-f006:**
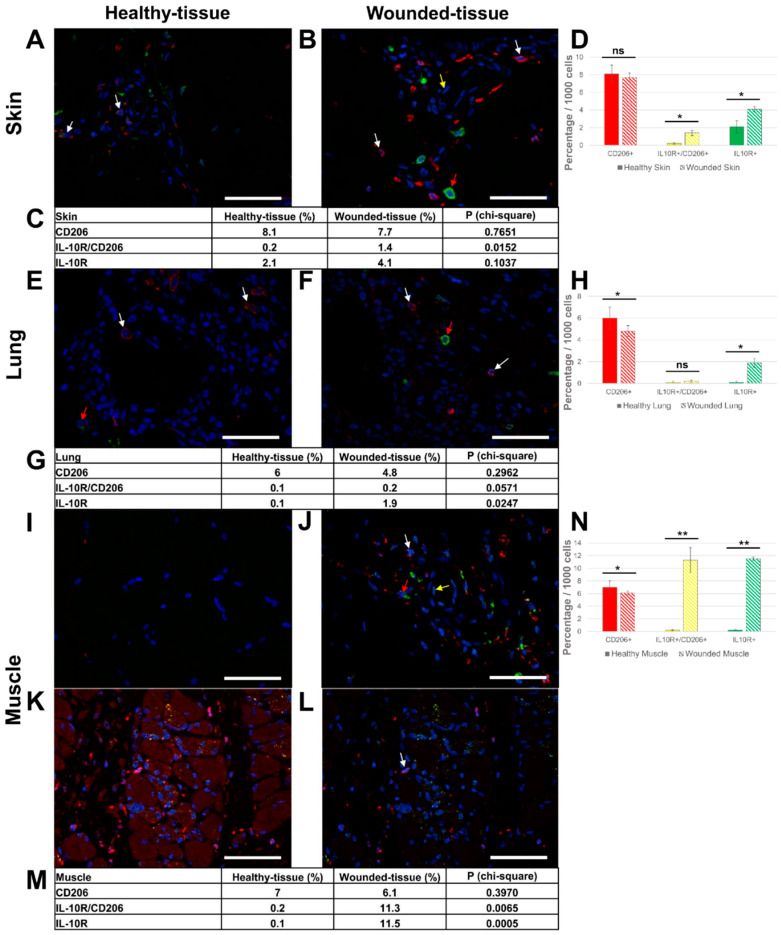
Identification of IL-10R+ and CD206+ cells in situ. (**A**,**B**) Immunolabeling of both IL-10R and CD206 in the skin, (**E**,**F**) the lung, and (**I**–**L**) the muscle sections. (**L**) identical to the K image, including the autofluorescence of muscle cells. (**A**,**E**,**I**) Healthy tissue. (**B**,**F**,**J**,**K**,**L**) Wounded tissue. (**C**,**D**) Quantitative analysis of the IL-10R+ and CD206+ cells in the skin; (**G,H**) the lung and (**M**,**N**) the muscle. Quantitative analysis based on random examination of 3 sets of 1000 cells per condition. Anti-IL-10R (Alexa488, green fluorescence) and Anti-CD206 (Alexa568, red fluorescence) label the tissue-resident and M2-like macrophages. Nuclear staining with DAPI (blue fluorescence). Scale bar = 50 µm. Yellow arrow: endothelial cell; white arrow: macrophage; and red arrow: IL-10R+ cell. ** *p* < 0.01, * *p* < 0.5, ns = not significant, compared to healthy tissue.

**Figure 7 cells-12-01522-f007:**
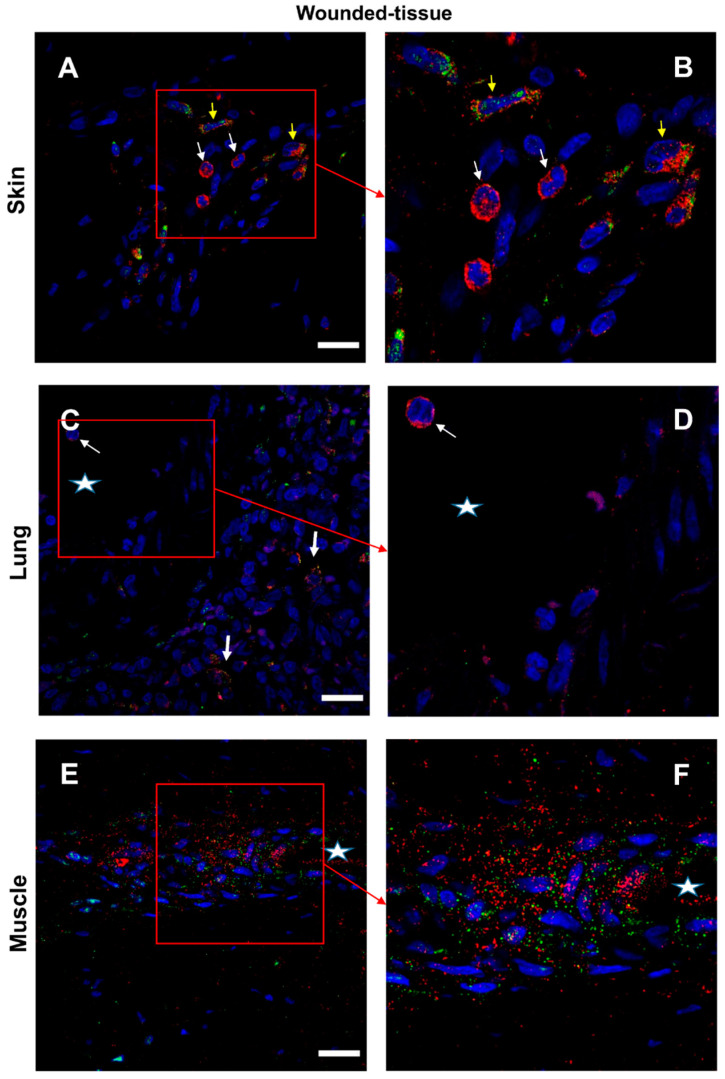
Identification of IL-4R+ and IL-4+cells in situ. (**A**,**B**) Immunolabeling of both IL-4R and IL-4 in the wounded skin, (**C**,**D**) the lung, and (**E**,**F**) the muscle sections. (**B**,**D**,**F**) Expanded view: high magnification image of the area within the red rectangle in A,C, and E, respectively. Anti-IL-4R (Alexa488, green fluorescence) and Anti-IL-4 (Alexa568, red fluorescence). Nuclear staining with DAPI (blue fluorescence). Scale bar = 20 µm. Star: lumen of the blood vessel; white arrow: IL-4+ granulocyte; and yellow arrow: IL-4R+/IL-4+ cell.

**Figure 8 cells-12-01522-f008:**
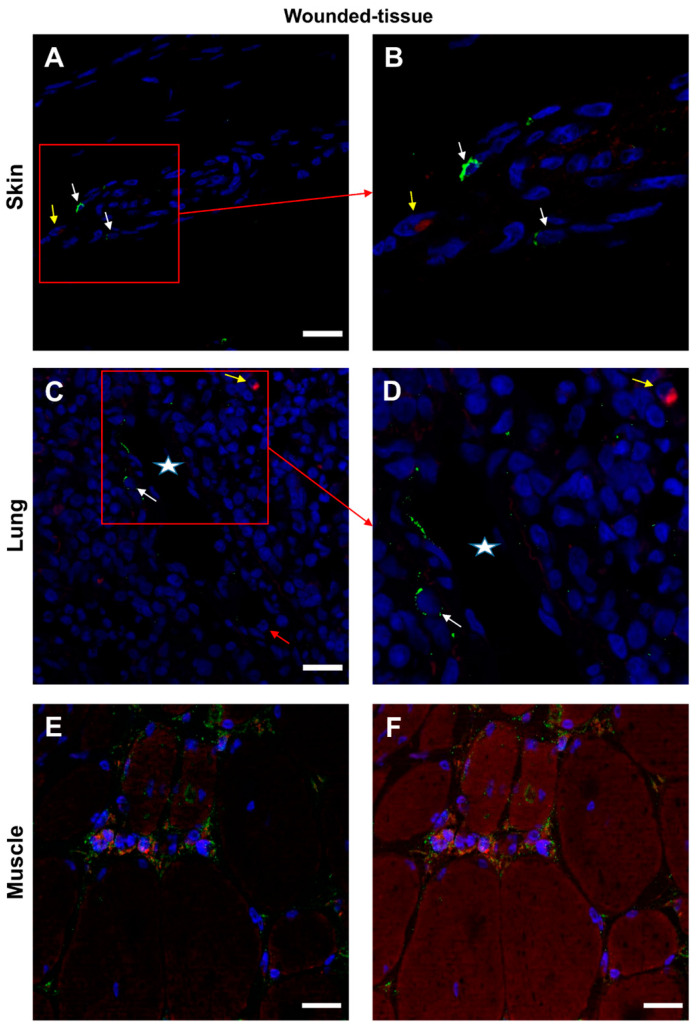
Identification of IL-10R+ and IL-10+ cells in situ. (**A**,**B**) Immunolabeling of both anti-IL-10R and IL-10 antibodies in the wounded skin, (**C**,**D**) the lung, and (**E**,**F**) the muscle sections. (**B**,**D**) Expanded view: high magnification image of the area within the red rectangle in (**A**,**C**), respectively. (**F**) Identical to the E image, including the autofluorescence of muscle cells. Anti-IL-10R (Alexa488, green fluorescence) and Anti-IL-10 (Alexa568, red fluorescence). Nuclear staining with DAPI (blue fluorescence). Scale bar = 20 µm. Star: lumen of the blood vessel; white arrow: IL-10R+ cell; yellow arrow: IL-10+ cell; and red arrow: granulocyte in the blood vessel.

**Figure 9 cells-12-01522-f009:**
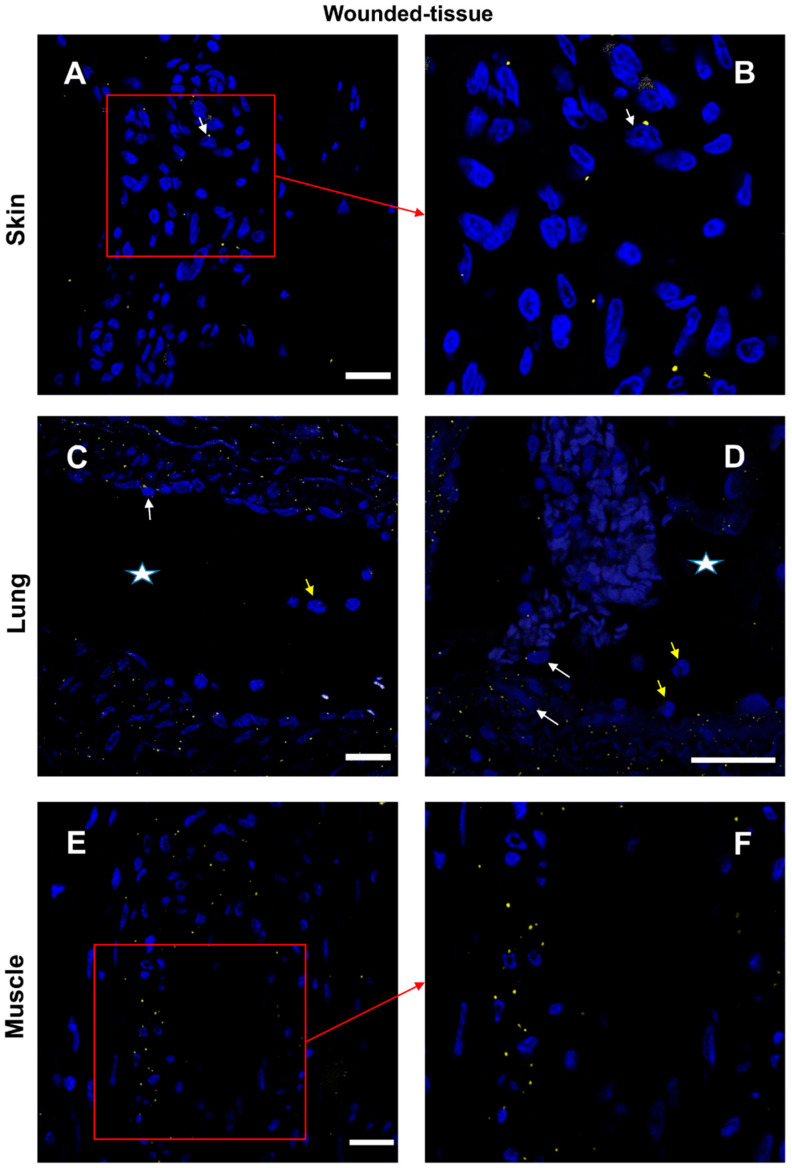
Interaction between IL-4 and IL-4R in situ. (**A**,**B**) Proximity Ligation Assay with anti-IL-4 and anti-IL-4R antibodies in the wounded skin, (**C**,**D**) the wounded lung, and (**E**,**F**) the wounded muscle. (**B**,**F**) Expanded view: high magnification image of the area within the red rectangle in A,E. Alexa568, yellow fluorescence, labeling the interaction between IL-4 and IL-4R. Nuclear staining with DAPI (blue fluorescence). Scale bar = 20 µm. Star: lumen of the blood vessel; white arrow: Proximity Ligation Assay IL-4R+/IL-4+ cells; and yellow arrow: granulocyte.

**Figure 10 cells-12-01522-f010:**
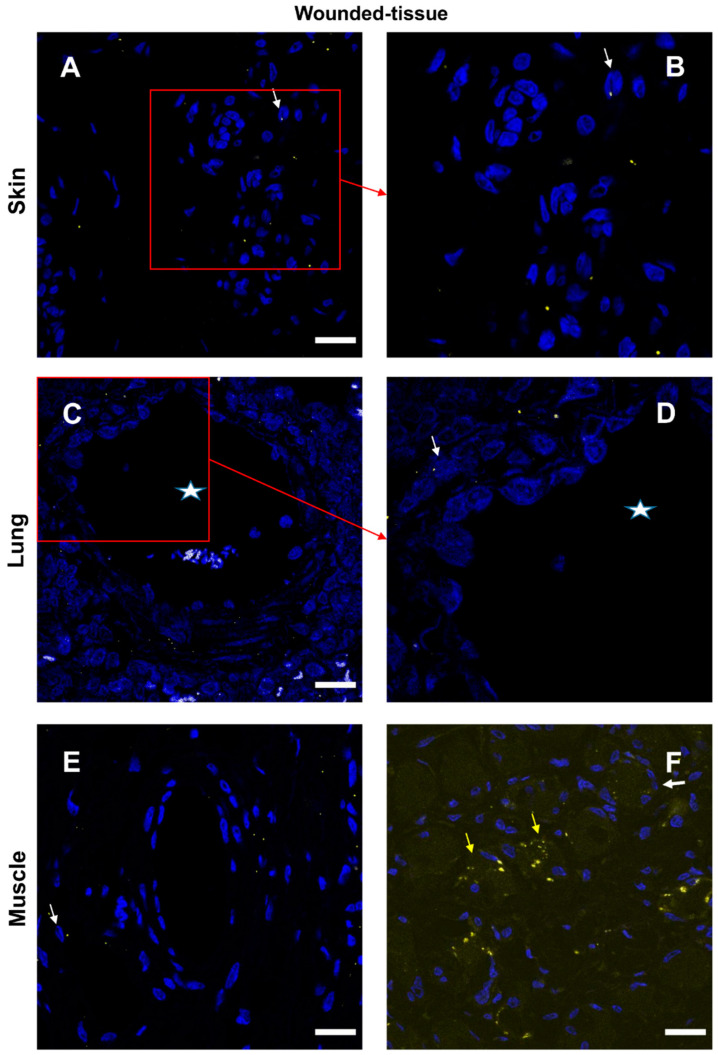
Interaction between IL-10 and IL-10R in situ. (**A**,**B**) Proximity Ligation Assay with anti-IL-10R and anti-IL-10 in the wounded skin (**C**,**D**) the wounded lung and (**E**,**F**) the wounded muscle. (**B**,**D**) Expanded view: high magnification image of the area within the red rectangle in (**A**,**C**). Alexa568, yellow fluorescence, labeling the interaction between IL-10 and IL-10R. Nuclear staining with DAPI (blue fluorescence). Scale bar = 20 µm. Star: lumen of the blood vessel; white arrow: Proximity Ligation Assay IL-10R/IL-10+ cells; and yellow arrow: muscle fiber.

**Figure 11 cells-12-01522-f011:**
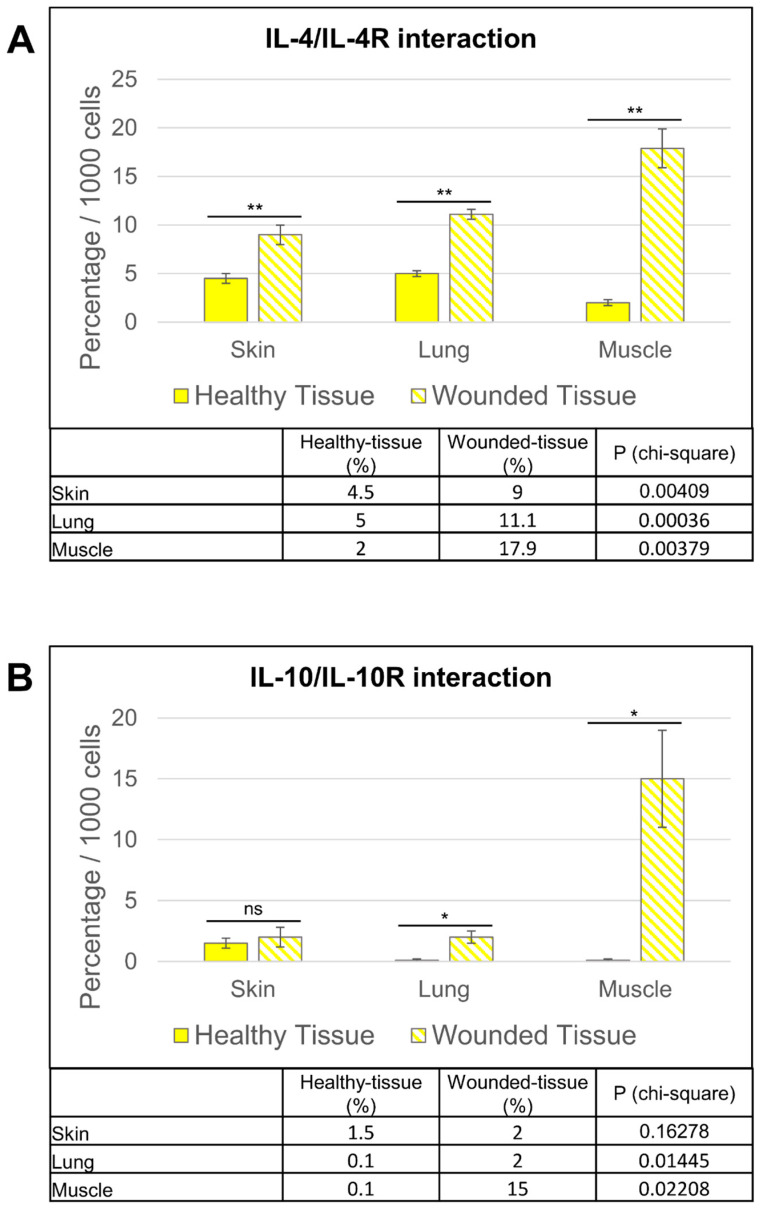
Quantitative analysis of both (**A**) IL-4/IL-4R and (**B**) IL-10/IL-10R interactions. Quantitative analysis based on random examination of 3 sets of 1000 cells per condition. ** *p* < 0.01, * *p* < 0.5, ns = not significant, compared to healthy tissue.

**Figure 12 cells-12-01522-f012:**
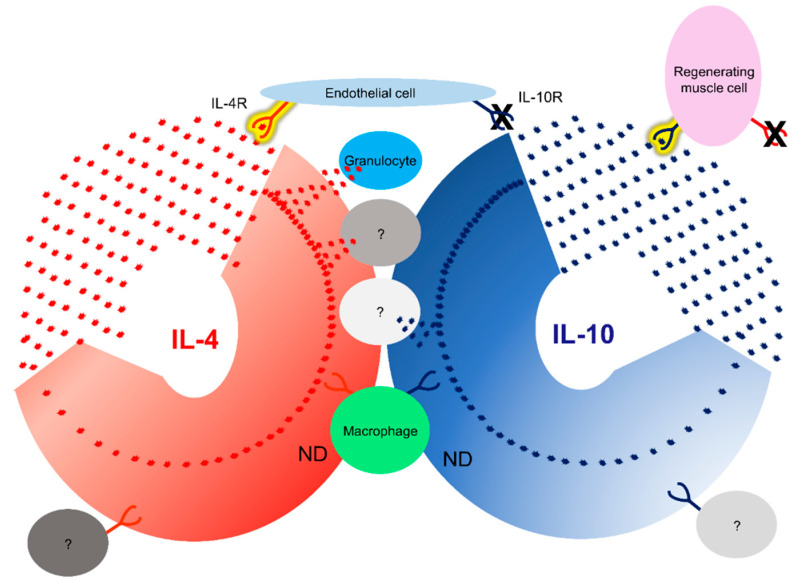
Summary of our results on the in situ identification of IL-4 and IL-10 cytokine receptor interactions. Endothelial cells showed intense IL-4 cytokine receptor interaction and muscle cells IL-10 cytokine receptor interaction during regeneration. The presence of IL-4 was detected in granulocytes and both receptors in macrophages. Contrary to the literature, the presence of IL-10R in endothelial cells and IL-4R in muscle cells was not detected. In addition, we identified further cells producing cytokines or their receptors (marked with a question mark), which will be characterized in the future. IL-4 is shown in red and IL-10 in blue. Interactions detected are highlighted in yellow.

**Table 1 cells-12-01522-t001:** Expression of the IL-4 and IL-10 and their specific receptors. IL-4 in red, IL-10 in blue.

Expression	IL-4		IL-10	
	Cytokine	Receptor	Cytokine	Receptor
Granulocyte	+	+	+	+
T cell	+	+	+	+
Mast cell	+	+	+	+
Macrophage	+	+	+	+
Monocyte	−	+	+	+
B cell	−	+	+	+
DC cell	−	+	+	+
NK cell	−	+	+	+
ILC2	−	−	−	−
Endothelial cell	−	+	−	+
Fibroblast	−	+	−	−
Muscle cell	−	+	−	−

**Table 2 cells-12-01522-t002:** Cytokine–receptor interaction for both IL-4 and IL-10 was identified in situ. (A) expression and (B) interaction during tissue regeneration. IL-4 in red, IL-10 in blue. *: was not identified in muscle tissue, because only a few granulocytes could be detected; ND: not determined.

**(A)**				
**Expression**	**IL-4**		**IL-10**	
Skin/Lung/Muscle	Cytokine	Receptor	Cytokine	Receptor
Granulocyte	+ *	−	−	−
Macrophage	ND	+	ND	+
Endothelial cell	−	+	−	-
Regenerating muscle cell	−	−	−	+
Uncharacterized cell	+	+	−	−
Uncharacterized cell	−	−	+	+
**(B)**				
**Cytokine/Receptor Interaction**	**IL-4/IL-4R**		**IL-10/IL-10R**	
Endothelial cell	+		−	
Regenerating muscle cell	−		+	

## Data Availability

Not applicable.

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
