# Peer review of "In Situ Identification of Both IL-4 and IL-10 Cytokine–Receptor Interactions during Tissue Regeneration"

_cells, 2023, doi:10.3390/cells12111522_

Round 1

Reviewer 1 Report

The paper submitted by Nikovics et al. about IL-4 and IL-10 cytokine receptors and the binding of the respective ligands in the context of tissue regeneration is in principle an interesting study. However, the paper lacks in most figures the quantification of the results obtained by PLA analyses, which makes it impossible for the reader/reviewer to interpret the representative immune-histo-pictures. The data provided in this study is more or less only descriptive and the authors offer no explanation why the find some receptor/ligand interactions on some cells and not on other which also express the receptors. Therefore, in its current form, the paper is not suitable for publication in a high quality journal.

Author Response

Reviewer #1:

Dear Reviewer 1,

The manuscript has been modified and comments sincerely helped to improve our work. We have improved the manuscript by taking into account each of the reviewers’ comments and as required, new Figure 1, Figure 3, Figure 4, Figure 5, Figure 6, Figure 7, Figure 8, Figure 9, Figure 10, and Figure 11 have been included. In all figures, arrowheads and bold arrows have been removed and replaced with arrows of a different color. In addition, all arrows have been better positioned (see Figures 3, 4, 5, 6, 7, 8, 9, 10) and arrows pointing to nowhere have been removed.

The paper submitted by Nikovics et al. about IL-4 and IL-10 cytokine receptors and the binding of the respective ligands in the context of tissue regeneration is in principle an interesting study. However, the paper lacks in most figures the quantification of the results obtained by PLA analyses, which makes it impossible for the reader/reviewer to interpret the representative immune-histo-pictures. The data provided in this study is more or less only descriptive and the authors offer no explanation why they find some receptor/ligand interactions on some cells and not on others that also express the receptors. Therefore, in its current form, the paper is not suitable for publication in a high quality journal.

Thank you for your comment. The comment has been taken into account. The additional quantification of the PLA results (Figure 11) has been added and the results and the discussion have been modified. (See lines 468-475, 525-527, 581-591, 602-609, 614-671, 675-702, 810-817).

In addition, to help readers to understand our strategy to validate a protein-protein interaction using the PLA method, the Figure 1B was added.

Reviewer 2 Report

In this study, the authors Nikovics et al used Proximity Ligation Assays to investigate the interaction between IL4 and IL10 with respective receptors IL4R and IL10R in a various tissues from pig animal model in situ. The study is innovative and interesting. However, some results are not convincing or rigorous. The writing also needs major improvement with better explanation of the results to make it appeal to a broader audience. 

1) Fig.1, 2 and 11, better description of the figure is needed. For example, what are the cells with question marks in Fig.1? There are so many things happening in Fig.2, a more detailed explanation/description is needed.  

2) The authors should perform some westernblotting to show the increase of IL4R and IL10R in the wounded tissue to make the data more convincing. 

3) In the PLA assay, the yellow fluorescence signal from PLA appear to be very low. Given the amount of cytokines released, this appears to be not conclusive enough. Some statistical analyses should be performed.  The authors should also include some positive control to make sure the experiments actually worked. 

4) What are differences between Fig. E and F? The yellow fluorescence in F is so much stronger than E. 

5) The discussion needs improvement. Right now it is mainly a summery of the results. 

Minor: 

1) line 175, H2O2, improper format

2) line 235, reference of CD206 marker should be provided

3) Fig. 2, The use of arrows and arrowheads are confusing. Some of the arrows need to be placed better, as they point to nothing. There are also some thick arrows.

4) Methods for the imaging in PLA are needed. 

The writing needs improvement. 

Author Response

Reviewer #2:

Dear Reviewer 2,

The manuscript has been modified and comments sincerely helped to improve our work. We have improved the manuscript by taking into account each of the reviewers’ comments and as required, new Figure 1, Figure 3, Figure 4, Figure 5, Figure 6, Figure 7, Figure 8, Figure 9, Figure 10, and Figure 11 have been included. In addition to the science, we paid careful attention to the English language and editing.

In this study, the authors Nikovics et al used Proximity Ligation Assays to investigate the interaction between IL4 and IL10 with respective receptors IL4R and IL10R in various tissues from pig animal model in situ. The study is innovative and interesting. However, some results are not convincing or rigorous. The writing also needs major improvement with better explanation of the results to make it appeal to a broader audience.

Thank you for your comment. The comment has been taken into account. The manuscript has been modified to clarify the paper. To improve the explanation of the results and to help readers to understand our strategy Figures 1B, 11 were added. (See lines 53-68, 468-475, 525-527, 581-591, 602-609, 614-671, 675-702, 810-817)

  1. Fig.1, 2, and 11, better description of the figure is needed. For example, what are the cells with question marks in Fig.1? There are so many things happening in Fig.2, a more detailed explanation/description is needed.

Thank you for your suggestion. Figures 1,2,12 have been explained in more detail. (See lines 53-68, 131-140, 614-661)

  1. The authors should perform some westernblotting to show the increase of IL4R and IL10R in the wounded tissue to make the data more convincing.

We are very sorry, but unfortunately, we currently do not have the tissue for western blotting. New pigs will not be available until the end of the year. There are only tissues embedded in paraffin.

  1. In the PLA assay, the yellow fluorescence signal from PLA appear to be very low. Given the amount of cytokines released, this appears to be not conclusive enough. Some statistical analyses should be performed. The authors should also include some positive control to make sure the experiments actually worked.

Thank you for your remark. The quantification has been performed and added. (see Figure 11 and lines 468-475, 525-527, 675-702)

Both cytokine-receptor interactions were more intense in damaged tissue than in healthy tissue. The percentage of IL-4/IL-4R interactions detected in lung and muscle was approximately equal to the percentage of cells expressing the receptor in both healthy and injured tissues. In skin, however, the percentage of these cytokine-receptor interactions was approximately half the percentage of cells expressing the receptor. Further research is needed to understand this difference. The analysis of the IL10 cytokine is even more complicated. In healthy tissues, the level of cytokine-receptor interaction was approximately equal to the number of cells expressing the receptor, but in damaged tissues, except lung tissue, fewer IL-10/IL-10R interactions were detected than the number of cells expressing the receptor.

In addition, only a few cytokine-receptor interactions per cell were observed for both cytokines in PLA assays. This phenomenon can be explained by the fact that the duration of the interaction between different cytokines and its receptor can vary depending on several factors, including the concentration of the cytokine, the availability of the receptor at the cell surface, and the downstream signaling events triggered by the interaction. In general, the cytokine-receptor interaction is a transient event and typically lasts for a short period to ensure dynamic and accurate signal transduction. The actual binding time between cytokine and receptor at the molecular level is typically in the range of microseconds to milliseconds. However, it is important to note that the total duration of cytokine-receptor interaction and subsequent signal transduction may be longer due to downstream signaling events and internalization of the receptor-ligand complex. Internalization may lead to the cessation of signal transduction and possible degradation or recycling of the receptor-ligand complex. In addition, the low number of cytokine-receptor interactions for IL-4 may be explained by the fact that it may bind to the receptor with another cytokine, unlike IL-10, which is the only ligand for the IL-10 receptor. It is important to emphasize that the exact duration of cytokine-receptor interaction and signal transduction is context-dependent and may vary in different cellular and physiological contexts, and further research is needed to fully understand the exact temporal aspects of the interaction.

Reviewer 2 asked us to use a positive control to make sure that our experiments worked well. The percentage of most cytokine-receptor interactions was about the same as the percentage of cells expressing the receptor. Therefore, we did not use a positive control, but if the reviewer insists, we certainly will.

  1. What are the differences between Fig. E and F? The yellow fluorescence in F is so much stronger than E.

The two images were taken in the same muscle section in different regions. Large differences in the IL-10/IL-10R interaction were observed in this sample, with stronger interactions observed in the regenerating muscle than around the blood vessels.

  1. The discussion needs improvement. Right now it is mainly a summery of the results.

The comment has been taken into account; the discussion and the conclusion have been modified. (See lines 581-591, 602-609, 614-671, 675-702, 810-817)

Minor:

  1. line 175, H2O2, improper format

H2O2 was modified to H2O2.. (See line 224)

  1. line 235, reference of CD206 marker should be provided

Thank you for your comment, the reference has been added. (See line 287)

  1. Fig. 2, The use of arrows and arrowheads are confusing. Some of the arrows need to be placed better, as they point to nothing. There are also some thick arrows.

Thank you for your comment. In all figures, arrowheads and bold arrows have been removed and replaced with arrows of a different color. In addition, all arrows have been better positioned (see Figures 3, 4, 5, 6, 7, 8, 9, 10) and arrows pointing to nowhere have been removed.

  1. Methods for the imaging in PLA are needed.

PLA imaging methods have been added. (See lines 261-262)

Round 2

Reviewer 1 Report

The revision has significantly improved the quality of the paper. So, it is now suitable for publication in Cells.

Reviewer 2 Report

The authors have sufficiently addressed my concerns and comments.